# An Outbreak of a Respiratory Disorder at a Russian Swine Farm Associated with the Co-Circulation of *PRRSV1* and *PRRSV2*

**DOI:** 10.3390/v12101169

**Published:** 2020-10-15

**Authors:** Sergei Raev, Anton Yuzhakov, Alexandr Bulgakov, Ludmila Kostina, Alexei Gerasianinov, Oleg Verkhovsky, Alexei Zaberezhny, Taras Aliper

**Affiliations:** 1Federal State Budget Scientific Institution “Federal Scientific Center VIEV”, 109428 Moscow, Russia; anton_oskol@mail.ru (A.Y.); bulgakov_ad@mail.ru (A.B.); lvkostina@mail.ru (L.K.); zaberezhny@mail.ru (A.Z.); coronavirus@yandex.ru (T.A.); 2Siberian Agrarian Group, 634009 Tomsk, Russia; gerasyanovav@sagro.ru; 3Laboratory of Virology, Diagnostics and Prevention Research Institute for Human and Animal Diseases, 123098 Moscow, Russia; info@dpri.ru

**Keywords:** *PRRSV1*, *PRRSV2*, *PCV2*, *ORF7*, cross-sectional, phylogenetic analysis, antibody detection

## Abstract

We conducted a cross-sectional study to identify the major respiratory pathogen responsible for an outbreak of respiratory disease at a swine farm in West Siberia in 2019. We discovered that the peak of morbidity and mortality coincided with a high level of porcine reproductive and respiratory syndrome virus (*PRRSV*) 1 and 2-related viremia. Based on longer PRRSV2 viremia, the dominant role of PRRSV2 over PRRSV1 in the outbreak was assumed. Phylogenetic analysis revealed that the PRRSV1 strain belonged to sub-genotype 2—one of the predominant groups of genotype 1 PRRSVs in Russia. A partial open reading frame 7 sequence of the PRRSV2 isolate demonstrated a high identity with modified live vaccine-related strains from Denmark (93%) and wild-type VR2332 (92%). We identified the first instance of *PRRSV1/PRRSV2* mixed infection in Russia. This finding indicates that further field investigations are needed to access *PRRSV2* epidemiology in eastern Europe.

## 1. Introduction

Porcine reproductive and respiratory syndrome (PRRS) has been simultaneously described in the USA and Europe and remains a major problem for the pig industry. Economic losses associated with this disease are estimated at US$664 million per year globally. In sows, reproductive failure is characterized by abortions, fetus mummification, stillbirths, and birth of weak offspring. Growing piglets mainly demonstrate respiratory symptoms such as dyspnea, coughing, and fever [1].

Further investigations have revealed that the causative agent of this disease is porcine reproductive and respiratory syndrome virus (PRRSV), for which two genotypes have been described: the *European* (PRRSV1) and *North American* (*PRRSV2*) genotypes. Significant genetic (50–60% identity) and antigenic differences eventually led to their divergence into two separate species: *Betaarterivirus suid 1* (the *European type*, or *PRRSV1*) and *Betaarterivirus suid 2* (the *North American* type, or *PRRSV2*) [2]. Both viruses belong to genus *Betaarterivirus* in the *Variartevirinae* subfamily and *Arteriviridae* family. *PRRSV*’s genome comprises about 15,000 nucleotides and contains ten open reading frames (ORFs) including *ORF7*, which encodes the nucleocapsid protein [3].

Shortly after PRRSV’s discovery, both *PRRSV1* and *PRRSV2* were isolated beyond their original places of detection, namely Europe and North America, respectively. *PRRSV1* strains are currently present in Europe, North America, and Asia, and *PRRSV2* strains are predominant in North America and Asia but commonly found in Europe. Since *PRRSV2* was first isolated in Denmark in 1995; similar (vaccine-related) strains were detected in Europe on numerous occasions. Despite wild-type PRRSV2 strains being isolated in Germany and Hungary, modified live vaccine (MLV)-related strains remain a major group of genotype 2 PRRS viruses circulating in Europe [4,5,6,7,8,9].

Co-circulation of *PRRSV1* and *PRRSV2* within one continent and the absence of cross-protection between them have logically resulted in dual infections at farms. For instance, both *PRRSV1* and PRRSV2 were found at 24.2% of infected farms in South Korea [10].

In Russia, PRRS is mainly caused by *PRRSV1-1* (including the so-called atypical Russian group of viruses) and *PRRSV1-2* [9,11,12]. Only limited data are available on *PRRSV2* epidemiology in Russia. One PRRS outbreak caused by highly pathogenic *PRRSV2* (*JXA1*-related) was described in 2007 [13]. Notably, *PRRSV2* is periodically detected on its own or simultaneously with PRRSV1 by veterinary diagnostic laboratories in Russia. However, due to the comparably low incidence of *PRRSV2* detection, it is not considered and often goes unnoticed. 

In August 2019, an outbreak of acute respiratory disease was recorded at a swine farm in Kemerovo region, Russia. The outbreak was mainly characterized by dispone, fever, and anorexia. According to the farm’s veterinary department data, morbidity and mortality (up to 40%) peaked in 7- to 12-week-old piglets. This 1500-sow farrow-to-finish farm was located more than 100 km away from another swine unit. The sow vaccination schedule included quarterly immunization with a killed vaccine against *PRRSV* (*Lelystad*-based) *porcine parvovirus*, *pseudorabies* virus, and leptospirosis. Piglets received a *porcine circovirus type 2* (*PCV2*) subunit vaccine and a killed vaccine (one dose each) and two doses of live classical swine fever vaccine. Based on the latest laboratory update at this farm in 2012, the farm was endemic for *PRRSV*, *PCV2*, and *Mycoplasma hyopneumoniae* (M.hyo). Considering the current outbreak and the long period of time that had passed since the last laboratory investigations were carried out, a decision was made to conduct a cross-sectional surveillance study.

The main goal of this investigation was to identify the major causative agent of respiratory outbreak at the farm and partially characterize the genome sequences of the isolated viruses.

## 2. Materials and Methods

### 2.1. Samples

The blood samples were taken from piglets ages 0, 3, 7, 12, 16, 20, and 26 weeks (8 to 10 animals per age group). Serum samples were collected on the same day and stored frozen at −70 °C before analysis.

### 2.2. ELISA

All serum samples were tested for the presence of antibodies against *PRRSV* capsid protein using two commercial ELISA test kits. The first kit, INgezim PRRS Universal (Ingenasa, Madrid Spain), is prepared on the basis of both *PRRSV1* and *PRRSV2* recombinant nucleoproteins. This kit detects antibodies against both *PRRSV1* and *PRRSV2* designated as “pan-*PRRSV* antibodies”. The second assay, RRSS-SEROTEST (Vetbiochem, Moscow, Russia), is capable of detecting antibodies against *PRRSV1* only. Both ELISA test kits were used in accordance with the manufacturers’ instructions. 

### 2.3. PCR and Sequencing

*PRRSV*, *PCV2*, *swine influenza virus* (*SIV*), *porcine parvovirus virus* (*PPV*), and *porcine respiratory coronavirus* (*PRCV*) in serum samples were detected using commercial polymerase chain reaction (PCR) kits (Vetbiochem, Russia) according to the manufacturer’s instructions. In all serum samples, detection was performed using commercial real-time polymerase chain reaction (PCR) kits (Vetbiochem, Russia) in accordance with the manufacturer’s instructions. According to previous data, both *ORF5* and *ORF7* sequences might be chosen for *PRRSV1* classification [8]. We used the following primers for the amplification and sequencing of *PRRSV1 ORF7*, kindly provided by Ivan Trus (Ghe University, Ghent, Belgium): 5′-TGGCCCCTGCCCAICACGT-3′ (PRRSV1-ORF7-F) and 5′-TGATCGCCCTAATTGAATAGGTGACT-3′ (PRRSV1-ORF7-R). For the amplification and sequencing of *PRRSV2 ORF7*, we used two pairs of nested primers. For the first round of PCR, we used the following primers: 5′-TTCTGGCCCCTGCCCAYC-3′ (F7) and 5′-CGCCCTAATTGAATAGGTGAC-3′ (R7). For the second round of PCR as well as for sequencing, we used 5’-CCAAATAACAACGGCAAG-3’ (AmerF7_12) and 5’-TCATGCTGAGGGTGATG-3’ (AmerR7_12). The expected size of PCR products of *PRRSV1 ORF7* and *PRRSV2 ORF7* was around 644 and 368 nucleotides, respectively. A one-tube, real-time RT-PCR kit (Alpha Ferment, Moscow, Russia) was used to perform reverse-transcription polymerase chain reaction (RT-PCR). The PCR amplification program consisted of 10 min at 50 °C, 5 min at 95 °C, 35 cycles at 95 °C for 15 s, 55 °C for 15 s, and 72 °C for 30 s, as well as 1 cycle of 5 min at 72 °C. The amplification products were visualized in 1% agarose gel in a buffer containing a mixture of Tris base, acetic acid and EDTA (x1 TAE) after which purification was performed using Monarch DNA Gel Extraction Kit (New England Biolabs, MA, USA) in accordance with the manufacturer’s instructions. Sanger sequencing was performed using the Big Dye® Terminator v.3.1 Cycle Sequencing Kit (Applied Biosystems, CA, USA) in accordance with the manufacturer’s instructions, and the primers specified above. The nucleotide sequences of the genome fragments were determined using an AB3130 genomic automated analyzer (Applied Biosystems, USA). The nucleotide sequences were analyzed using Lasergene 11.1.0. (DNASTAR, WI, USA). Multiple alignment was performed using MUSCLE (MEGA 7.0.18). Phylogenetic dendrograms were plotted using the maximum likelihood method, GTR model (MEGA 7.0.18). The topology of the trees was confirmed following 1000 bootstrapping replication steps.

## 3. Results

### 3.1. Viremial ELISA

Viral RNA was not detected in sera from either newborn or 3-week-old piglets, whereas in 7-week-old piglets, *PRRSV1* and *PRRSV2*, and viral RNA were present in 50% and 40% of serum samples, respectively (Figure 1). The mixed RNA belonging to both viruses was detected in only one 7-week-old piglet. In sharp contrast to *PRRSV1*, where the RNA was not present in samples from 12-week-old animals, *PRRSV2* viremia was detected in 60% of 12-week-old piglets. Simultaneously, *PCV2* DNA was detected in 20% of 12-week-old piglets, and viremia rates increased with age until the age of 20 weeks, with 40% of the piglets tested being viremic. All serum samples were negative for *SIV*, *PPV*, and *PRCV*.

Whereas the total concentration of *PRRSV1*-specific antibodies showed a clear tendency to rise from 3 weeks (when all samples were seronegative) to 20 weeks of age, a minor numerical decrease in the total level of antibodies against both strains from 3 to 7 weeks of age was detected (Figure 2).

Comprehensive analysis of total *PRRSV* antibodies (to both genotypes) and *PRRSV1*-specific antibodies (Table 1) did not reveal any certain tendency. *PRRSV2* RNA was found in an only numerically higher (three out of nine) proportion of *PRRSV1* IgG negative/total *PRRSV* Ig-positive samples compared to *PRRSV1* (one out of nine).

### 3.2. Sequencing Results

Serum samples from 7-week-old piglets were used for *PRRSV1* (sample #4) and for *PRRSV2* (sample #6) sequencing. The *ORF7* sequences were deposited into the GenBank sequence database under the accession numbers *ORF7_Kem19EU-MT344126* and *ORF7_Kem19NA-MT344127*. The *ORF7_Kem19EU-MT344126* (*PRRSV1*) has 378 nucleotides, and further phylogenetic analysis revealed that this isolate belongs to subtype 2 of *PRRSV1* (Figure 3). This strain is most closely related to strains Eig и Aus from Lithuania and strain Gk from the European part of Russia [8].

The partial nucleotide sequence of *ORF7_Kem19NA-MT344127* (*PRRSV2*) comprises 337 nucleotides (Figure 4). The comparative analysis of partial sequence *ORF7_Kem19NA-MT344127* against those of reference strains demonstrated a high nucleotide identity with wild-type *VR2332* (92%) and MLV-related strains from Denmark: *KC577602.1_DK-2004-3-1* and *AF095479.1_strain_17704A* (93%) [14]. The level of identity between the Hungarian non-MLV-related strain *KM514315.1:14861-15232_Hungary_102_2012* and *ORF7_Kem19NA-MT344127* was only 81%. Nucleotide identity with *GU256774.1*, the only one *PRRSV2* strain detected in Russia, was 88%. 

Further genetic analysis (Figure 5a) revealed that *Kem19NA* had an amino acid insertion of lysine (K) at position 47 in the *nuclear localization signal* (*NLS*) sequence of the N protein. Similar amino acid insertions are present in *PRRSV1-2* strains (Figure 5b), which is remarkable given the debate on the evolution of *PRRSV1* and *PRRSV2*.

## 4. Discussion

The results of this study indicate that the respiratory disease outbreak correlated with increased *PRRSV*-related viremia rates, while it did not correlate with the prevalence of another major respiratory pathogen, *PCV2*. Remarkably, *PRRSV2*-related viremia peaked in a more pronounced manner and lasted longer compared to *PRRSV1*-related viremia. This might be an indication of the predominant role of *PRRSV2* in respiratory outbreak development. 

We assume that whereas total *PRRSV* antibodies from piglets aged 12 weeks might be considered as post-infectious, at 3 weeks of age, these antibodies are of maternal origin. Due to differences in sensitivity and specificity between the test kits used in this study and simultaneous *PRRSV1* and *PRRSV2* viremia detection, our effort to distinguish between PRRSV1 and PRRSV2 antibodies by using universal (*PRRSV1/PRRSV2*) and *PRRSV1*-specific ELISA test kits cannot describe the real situation. After the initial introduction of *PRRSV2* into Europe, differentiation between antibodies against *PRRSV* of different genotypes became crucial to *PRRSV* diagnosis. An ELISA assay for simultaneous detection and differentiation between antibodies to *PRRSV1* and *PRRSV2* was developed [15], but no such ELISA test kit is commercially available in Russia. Hence, the use of universal PCR test kits for simultaneous detection of both viruses is the only option to prevent *PRRSV2* from remaining undetected. 

Data on the comparative pathogenicity of *PRRSV1*, *PRRSV2,* and mixed infection are scarce. In one study, the predominant role of PRRSV2 over PRRSV1 in the course of *PRRSV1-1/PRRSV2* (lineage 1) mixed infection was demonstrated on the basis of replication and pathogenicity levels [16]. Significant differences in the pathogenicity of different genotypes/lineages of PRRSV1 and PRRSV2 should be considered in the case of mixed infection. Further experimental studies will be required to establish the impact of at least *PRRSV1-2* and *PRRSV2* (lineage 5) presence. Despite recombination never being demonstrated between *PRRSV1* and *PRRSV2* strains, the simultaneous presence of *PRRSV1* and *PRRSV2* might also be considered a possible risk for the emergence of recombinant strains. The use of MLV vaccines under such circumstances may only increase this risk [17,18].

The idea that *PRRSV1* is a homogenous group of viruses was revised after the discovery of genetically distinct isolates from eastern Europe, especially from Belarus and Russia [9]. Thus, the detection of a new member of *PRRSV1-2* in the present study offers additional proof of the wide distribution of this genotype in Russia. Isolates of *PRRSV1-1* (including the so-called Russian group of viruses), *PRRSV1-2*, and *PRRSV1-3* differ significantly in pathogenicity [11,12,19,20]. Marked differences in the nucleotide sequence between *PRRSV1-2* isolates and *PRRSV1* vaccine strains, which all belong to PRRSV1-1, including differences in the ORF5 sequence (which encodes a major epitope: the envelope glycoprotein), could potentially hamper vaccine efficacy [12]. The benefits of using currently available vaccines against *PRRSV1-2*, be they live, inactivated, or subunit, remain unknown. The circulation of *PRRSV2* further complicates the search for working control strategies, as no vaccines against *PRRSV2* are currently available in Russia. 

Phylogenetic analysis of a partial *ORF7* sequence of the *PRRSV2* strain isolated in the course of this study revealed that this strain was very dissimilar (88% identity) to the only known *PRRSV2* strain (*JXA1*-related, sublineage 8.7) detected in East Siberia, Russia, in 2007. This suggests that a different instance of *PRRSV2* introduction into Russia occurred at a certain point in addition to the one resulting in the outbreak of 2007. Further genetic analysis (including an *ORF5* sequence) should be performed to classify this isolate more precisely.

High homology within lineage 5 of *PRRSV2* makes the distinction between the Ingelvac PRRS MLV vaccine strain, its parental strain *VR-2332, VR-2332*-related wild-type isolates, and vaccine-derived isolates very complicated and requires at least *ORF5* sequence analysis [18]. The Boehringer MLV vaccine has not been licensed in Russia. However, there are known cases of the detection of an MLV-like *PRRSV1* strain in Vietnam, regardless of the corresponding type of vaccine not being licensed in that country [21]. Notably, one locally-produced MLV vaccine against *PRRSV2* has been licensed but is currently commercially unavailable in Russia, and the sequence of the vaccine virus has not yet been published. It is also conceivable that the virus originated from breeder animals infected with an MLV-related *PRRSV2* strain. It is well known that MLV-related strains (including strains that have a high identity level with *ORF7_Kem19NA-MT344127*) are widely spread in Denmark, one of the major exporters of breeding pigs [14]. For example, the average number of pigs imported from Denmark to Russia has been more than 3000 animals a year over the past three years. Finally, an independent introduction of a wild-type *VR2332*-related strain could result in the emergence of the *ORF7_Kem19NA-MT344127* virus.

In the course of infection, *NLS* interacts with nuclear transport proteins, leading to the penetration of the N protein into the nucleus and nucleolus of infected cells [22]. A critical role of this element of the N protein in pathogenesis has been shown [23]. A similar insertion is known to be found in a Mexican field isolate *DMZC144/2016* (GenBank ID MF352184.1) [24]. This might indicate that both *Kem19NA* and *DMZC144/2016* have a nucleoprotein consisting of 124 amino acids in length. Complete ORF7 sequencing should be performed to confirm this assumption. Interestingly, *Kem19NA* and *DMZC144/2016* are genetically distant from each other, their nucleotide identity being only 87%, suggesting that this insertion may not be lineage-specific. 

## 5. Conclusions

To the best of our knowledge, this is the first report of *PRRSV1/PRRSV2* co-circulation in Russia. Considered alongside data from field veterinary laboratories, our observations indicate that PRRSV2 circulates in both the European and Asian parts of the country. The prevalence and distribution of *PRRSV2* in Russia therefore warrants further investigation.

## Figures and Tables

**Figure 1 viruses-12-01169-f001:**
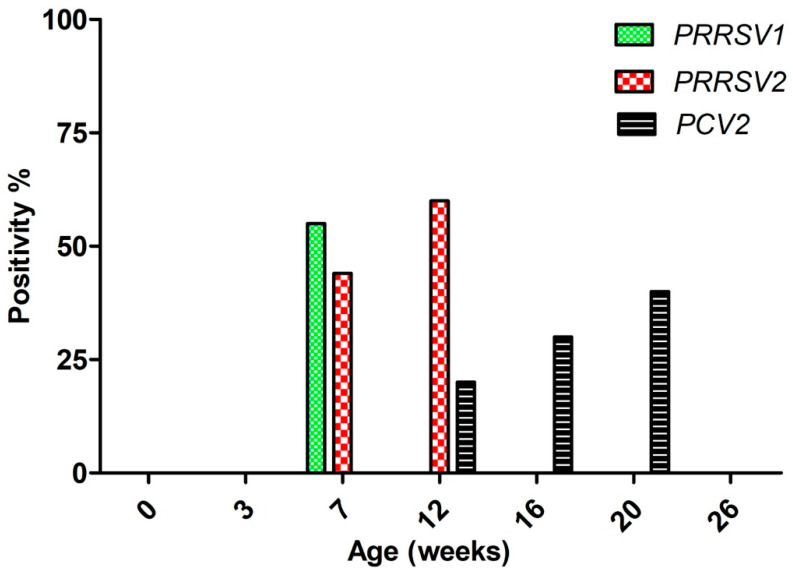
Viremia evaluation of *PRRSV1*, *PRRSV2*, and *PCV2* in different age groups.

**Figure 2 viruses-12-01169-f002:**
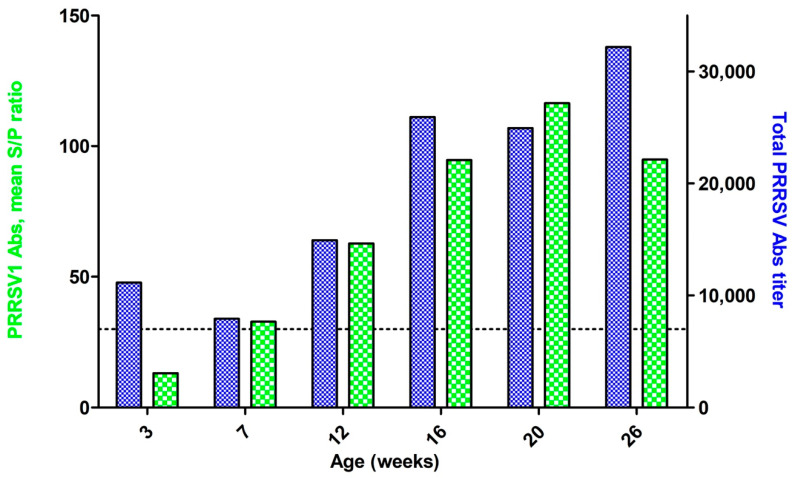
Anti-*PRRSV1* (green) and pan-*PRRSV* (blue) antibody detection.

**Figure 3 viruses-12-01169-f003:**
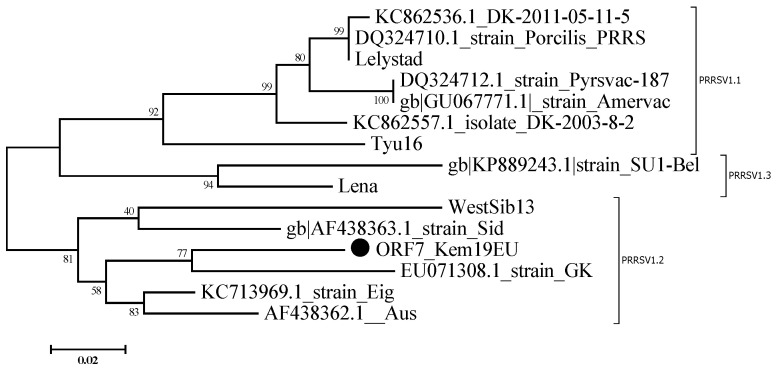
Phylogenetic trees of *PRRSV1* strains based on complete *ORF7* nucleotide sequences. Bootstrap confidence limits are presented at each node. The *Kem19EU* strain is designated by a circle (●). Strains are designated in the following format: GenBank accession number_ name.

**Figure 4 viruses-12-01169-f004:**
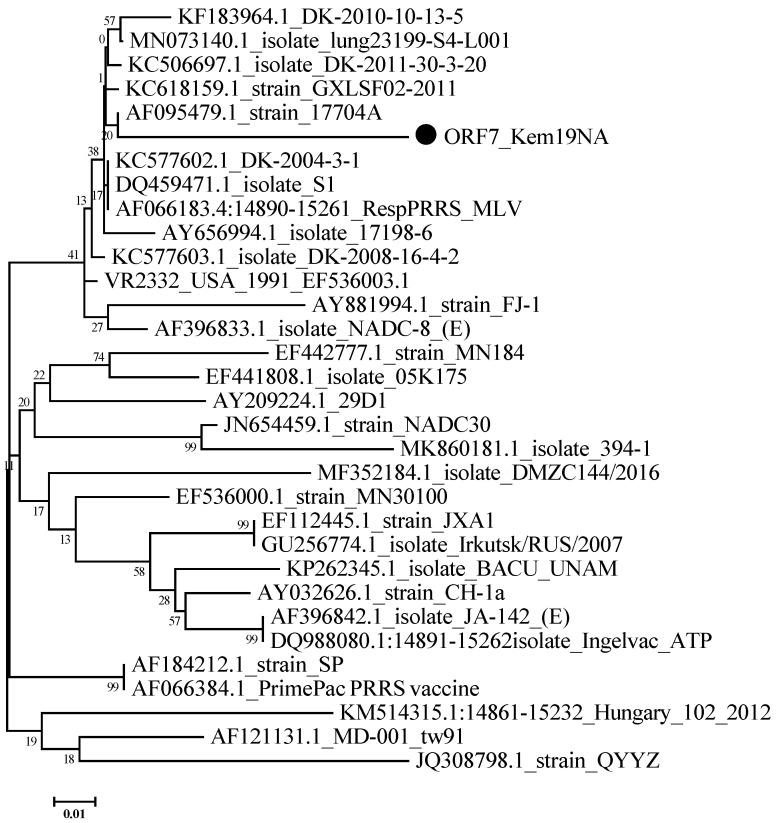
Phylogenetic trees of *PRRSV2* strains based on partial *ORF7* nucleotide sequences. Bootstrap confidence limits are presented at each node. The *Kem19NA* strain is designated by a circle (●). Strains are designated in the following format: GenBank accession number_ name.

**Figure 5 viruses-12-01169-f005:**
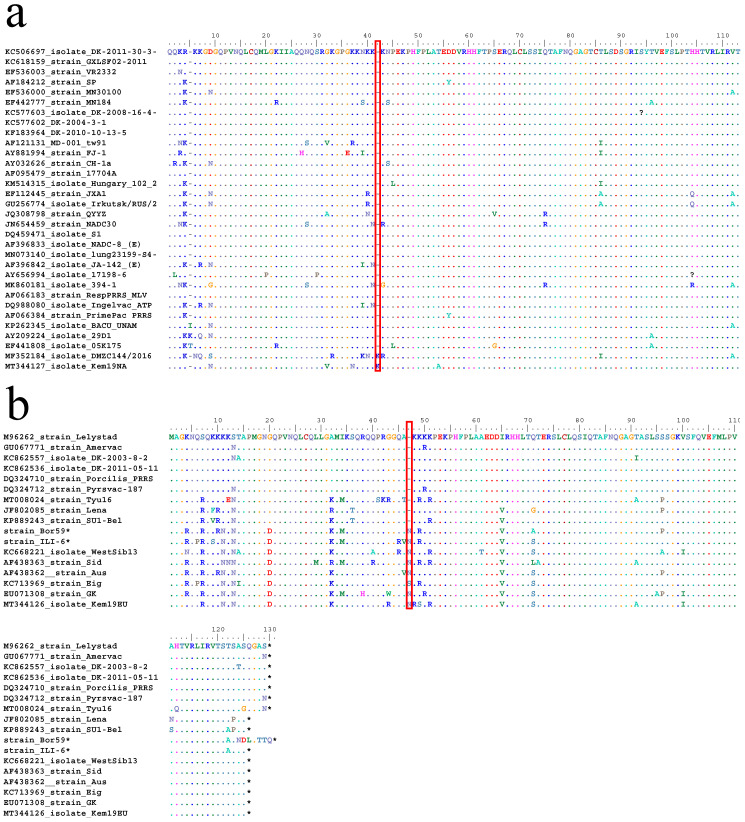
Partial alignment of amino acid sequences derived from *ORF7* of *PRRSV2* (**a**) and complete alignment of amino acid sequences derived from *ORF7* of *PRRSV1* (**b**) strains. Dots and hyphens represent identical amino acid positions and gapped positions, respectively. The red box denotes insertions in the investigated strains. The sequences of the strains indicated by an asterisk (*****) were kindly provided by Prof. Dr. Tomasz Stadejek (Warsaw University of Life Sciences, Poland) and Dr. Sara Botti (Parco Tecnologico Padano, Italy).

**Table 1 viruses-12-01169-t001:** Antibody detection. Detection of antibodies to *PRRSV1* (positive samples highlighted in green), antibodies to both, *PRRSV1* and *PRRSV2* (positive samples highlighted in blue), RNA from *PRRSV1* (positive samples highlighted in green), and RNA from *PRRSV2* (positive samples highlighted in red).

Week	Animal #	Antibody Detection (ELISA)	Viremia Detection (PCR)
*PRRSV1*	Pan-*PRRSV*	*PRRSV1*	*PRRSV2*
3	1				
2				
3				
4				
5				
6				
7				
8				
7	1				
2				
3				
4				
5				
6				
7				
8				
9				
12	1				
2				
3				
4				
5				
6				
7				
8				
9				
10				
16	1				
2				
3				
4				
5				
6				
7				
8				
9				
10				
20	1				
2				
3				
4				
5				
6				
7				
8				
9				
10				
26	1				
2				
3				
4				
5				
6				
7				
8

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
