# Peer review of "An Outbreak of a Respiratory Disorder at a Russian Swine Farm Associated with the Co-Circulation of PRRSV1 and PRRSV2"

_viruses, 2020, doi:10.3390/v12101169_

Round 1

Reviewer 1 Report

The paper describes the simultaneous detection of two different PRRSV species in samples from the same farm. The characterization of the viruses detected was done and  its potential role in the clinical disease observed in the farm is discussed.

The experimental design description and material and methods section should be reviewed. Specifically, a deeper description of the farm (location related to other farms, size, kind of operation, vaccination Schedule, endemic diseases) and of the clinical case should be included. It is also missing a description of the analysis done in the samples taken, and if all samples were used for all the techniques later described. The targets of some of the PCRs are also missing.

It is also unclear how many samples were used for sequencing analysis, since from the total of samples analyzed only two sequences have been provided. It should be described to which samples belong the sequences provided.

The authors should explain and add to the text why ORF7 was chosen for sequencing purposes, when it is widely accepted that ORF5 is the most adequate sample to perform phylogenic analysis and sequence homology comparisons.

The authors conclude that PCV-2 was not related to the clinical disease based only in the timing of detection of PCV-2 viremia. However, it is well known that porcine circovirus disease diagnostic cannot be only based in PCV-2 detection by PCR. Histopathology and IHC/ISH should be performed to rule out its implication.

The reviewer is missing a deeper investigation on the on the potential causes of the disease observed in the farm. The authors describe the analysis done for PCV-2 and PRRSV, but other pathogens such as Mycoplasma hyopneumoniae, PRCV, Actinobacillus pleuropneumoniae… should have been included as differential diagnosis. Please provide.

The authors discuss the role of maternal immunity in the timing of PRRSV viremia and antibodies. However, it is unclear which is the origin of the sow herd immunity. Please provide.

The reviewer disagrees with the conclusión of lines 176-177 about recombination events. Recombination has never been demonstrated between PRRSV-1 and PRRSV-2 strains and this should be clearly stated in the discussion. In this case vaccination could reduce the incidence of viremia and the risk of recombination events.

Other minor ítems:

  • There is a mistake in lines 44-45: PRRSV-1 and PRRSV-2 are quoted inversely.
  • Figure 1 could be more visual if bars were used instead of lines (suggestion)
  • Review indentation of lines 117 and 123
  • Review English style

Reviewer 2 Report

Summary

The authors reported the first case of PRRSV-1/PRRSV-2 mixed infection at a Russia swine farm based on a series of detection including ELISA, PCR for ORF7, and sequencing analysis. Although the cause of the epidemic is unknown, it has local epidemic characteristics, indicating that PRRSV-2 might be an epidemic viral strain in Eastern Europe.

Minor comment

Figure 1, Based on the description of samples in Methods, Line graph should not be used here because this is not a dynamic change, authors used different groups of animal at different week age, so cannot describe or conclude that “cases rise from 3 week” in figure lagend and text because of missing the data from 3 to 7 weeks.

Reviewer 3 Report

Authors described an outbreak of respiratory disease at a swine farm in West Siberia in 2019. The analysis revealed that animals were co-infected with PRRSV-1 and PRRSV-2 which is for the first time identified the PRRSV-1/PRRSV-2 mixed infection at pig farm in Russia. Based on longer PRRSV-2 viremia, the dominant role of PRRSV-2 over PRRSV-1 in the outbreak was assumed. Genetic typing in ORF7 revealed that PRRSV-1 belonged to subgenotype 2, PRRSV-2 was close to MLV strains from Denmark.

No doubt that this work extends our knowledge on epidemiology of PRRSV in Russia. It provides interesting results but manuscript should be prepared more carefully:

  • While the title uses PRRSV1 and PRRSV2, PRRSV-1 and PRRSV-2 are used in the text. Please clarify.
  • The farms where an outbreak was identified should be described in more details: size, age composition, vaccination, import/export…
  • Primers were provided by Ivan Trus - his affiliation is missing
  • We have no idea what size of PCR products were obtained by PCR using different set of primers.
  • Section Results should be written more carefully: Data from Fig. 2 and Table 1 are not mentioned in the text, only in legends, similar as Fig. 5 which is partially mentioned in Discussion only. If they are not so significant why to present them in manuscript?
  • The isolate Kem19NA has longer branch than other isolates (Fig. 4). This is usually observed when nucleotide sequence analysed is longer or shorter than others (after alignment) or chromatogram was not clearly readable. Please check if it is not your case.
  • Sequences are homologous or non-homologous (not with percentage). The percentage of identity (nt) or similarity (aa) should be used in the context of manuscript evaluated. Please correct.
  • Last paragraph of Discussion should be relocated to the first part of Discussion (after viremia part).
  • Reference are not unified; some journals are with abbreviations, others with full names.
  • Abstract: Twice is mentioned that further research is needed (lines 19 and 24) - abstract provides solid experimental data only. Opinions may be expressed in Discussion
  • It is really pity that authors did not sequence entire ORF7 and did not analyse the isolates in ORF5 region too. In this case the manuscript would be much stronger.

Generally: More attention is needed when scientific manuscript is released in the final form!

Round 2

Reviewer 1 Report

Accepted after review.